# The Vagueness of Vagueness in Noun Phrases

**Pierre-Henri Paris**                                PIERRE-HENRI.PARIS@TELECOM-PARIS.FR
**Syrine El Aoud**                                        SYRINE.ELAOUD@TELECOM-PARIS.FR
**Fabian Suchanek**                                  FABIAN.SUCHANEK@TELECOM-PARIS.FR
*Télécom Paris, Institut Polytechnique de Paris, France*

## Abstract

Natural language text has a great potential to feed knowledge bases. However, natural language is not always precise – and sometimes intentionally so. In this position paper, we study vagueness in noun phrases. We manually analyze the frequency of vague noun phrases in a Wikipedia corpus, and find that 1/4 of noun phrases exhibit some form of vagueness. We report on their nature and propose a categorization. We then conduct a literature review and present different definitions of vagueness, and different existing methods to deal with the detection and modeling of vagueness. We find that, despite its frequency, vagueness has not yet be addressed in its entirety.

## 1. Introduction

Natural language can be vague in many different ways: many verbs admit a wide range of interpretations (consider, e.g., "to work", which can mean a paid activity as well as a serious effort – and not always simultaneously). Noun phrases can be vague, too (as in "several countries"). Sometimes entire sentences do not have a clear-cut truth value (e.g., if they start with "In many cases,..."). Naturally, such phenomena can hinder the interpretation of natural language by automated means, e.g., in natural language understanding tasks, automated dialog systems, or intelligent assistants. Vagueness also poses a challenge for knowledge extraction. Consider for example this (condensed) excerpt from a featured article on Wikipedia:

> "An anti-tobacco sentiment grew in many nations from the middle of the 19th century."

This sentence is clearly factual and thus interesting for knowledge extraction. Yet, all noun phrases in the example are vague: what constitutes a "sentiment"? How many nations are "many"? Does the "middle of the 19th century" cover the 1930's? Thus, if we want to use this sentence for the construction of a knowledge base (KB), we face at least 3 challenges: **How can we detect vague entities?** Current information extraction approaches for KB construction often focus on named entities, and do not deal with phrases such as "anti-tobacco sentiments". **How can we model vague entities in a KB?** Current large-scale KBs cannot make statements about sets of unknown size such as "many nations". **How can we reason on vague entities?** If the sentiments grew in "many nations", we can deduce that there were more than "in few nations", but current KBs cannot do that type of reasoning.

We can, of course, choose to suppress vagueness, e.g., by reducing "many nations" to a single placeholder entity of type "nation", or by replacing it by a concrete number. However, this may not always be possible. It may not even always be desirable. Vagueness avoids being overly specific, when more information is not known or irrelevant at this point [Van Deemter, 2012, Lim and Wu, 2018]. Thus, vagueness is an important feature of natural language. Yet, it makes the sentence inaccessible to current information extraction methods for building KBs.

In this position paper, we shed light on the phenomenon of vagueness in noun phrases. We first review different definitions of vagueness in the literature, and propose a categorization (Section 2). Then, in Section 3, we study a corpus that was written with the intention to be least vague: a collection of featured Wikipedia articles. Our work is thus the first to actually quantify and categorize vagueness on real Web corpora. We find that 1/4 of noun phrases exhibit some type of vagueness. Finally, we discuss how vagueness has been treated in existing works. We find several works to detect vagueness in application-specific scenarios, such as privacy policies, requirement specifications, or ontologies (Section 4). In Section 5, we discuss how vagueness can be modeled from logical and probabilistic perspectives. Section 6 describes how all of these approaches could be put together. However, we also find that vagueness in general has not yet been addressed well in the literature. It remains, until now, a vague concept.

## 2. Defining Vagueness

**Definition.**    We concentrate here on the vagueness of noun phrases. This type of vagueness has been studied extensively in philosophy [Sorensen, 2018]. One classical illustration is the Heap Paradox, which is attributed to the Greek philosopher Eubulides of Miletus: A single grain of sand is not a heap. Nor does adding one more grain to some collection of sand make that collection a heap. But when does a collection of grains become a heap then? The Old Norse tale of Loki even suggests that any concept can be debatable and vague: When two executioners want to cut off Loki's head, Loki confuses them with a debate over where his head actually starts – thereby avoiding his execution. Thus, it appears that vagueness itself is a vague concept. We found two criteria that can help identify vagueness: One is the definition that a concept is vague if it admits borderline cases [Alexopoulos and Pavlopoulos, 2014, Liu et al., 2016]. Another is to take inspiration from the vagueness doctrine in American constitutional law: a concept is vague if it would not be admissible in a law. Both criteria are imperfect, but they make the "heap" vague, and the "head" not.

**Existing Categorizations.**    Several works have proposed categorizations of vague noun phrases. Liu et al. [2016] study vagueness at the sentence level and at the noun phrase level. They identify vagueness at the sentence level along the dimensions of condition ("as needed"), generalization ("generally"), and modality ("might"), and at the noun phrase level in the dimension of quantity ("some", "many"). Lassiter and Goodman [2017] distinguish two kinds of vagueness that can apply to noun phrases: scalar vagueness, where some expression can hold to some degree (with adjectives such as "tall", or modifiers such as "many"), and non-scalar vagueness, which they see in words such as "bird" and "fruit" (presumably because of a lack of more precision). Bennett [2005] identifies 3 types of vagueness: Threshold (where a property can hold to some degree), Partiality (where a certain proportion of a whole is concerned) and Deep Ambiguity (where a word regroups several disjoint but overlapping concepts).

We first tried to annotate a corpus (see Section 3) with these definitions from previous work. However, these definitions proved to be too abstract for our manual annotation of natural language text. For example, the concept of "beauty" is clearly vague. Yet, it is neither a vagueness of quantity (as in [Liu et al., 2016]), nor is it comparable in vagueness to a "bird" (as in [Lassiter and Goodman, 2017]), nor would we characterize it as an overlapping of disjoint but overlapping concepts (as in [Bennett, 2005]). The same applied to other vague noun phrases, either because they were carrying an opinion (like "beauty", "sophistication" or "fame"), or because they were hard to assess objectively (like "power" or "great ability"). Therefore, we adapted and completed these categories as follows.

**Proposed Categorization.**    We propose three categories of vagueness.

**Scalar vagueness** appears in expressions that can be interpreted as a scalar that ranges over a numerical scale and for which an unspecified threshold gives a truth value. For example, in "Mary is a *tall woman*", the height of Mary could range from 40 cm to 220 cm, and the assertion is true if Mary is taller than a certain implicit threshold (e.g., 180 cm). This type of vagueness also appears in [Rosadini et al., 2017, Bennett, 2005] and in the Heap Paradox.

**Quantitative vagueness** appears in expressions that refer to an unspecified portion of an entity (as in "a part of the film"), or to a set of entities whose number is not identified (as in "many scientists", or just "scientists"). This concept includes the numerical vagueness of [Liu et al., 2016, Bennett, 2005]. It applies generally to plural nouns without a definite article or cardinal, except if they refer to the concept itself (as in "Apples are fruits").

**Subjective vagueness** appears in expressions that can apply to a certain degree, and where there is no consensus on how to measure this degree. These often carry a subjective valuation, and correspond to Walter Bryce Gallie's "essentially contested concepts" [Gallie, 1955]. They include nouns such as "beauty" or "ingenuity" and echo the verse of the poet Fernando Pessoa: "*To define the beautiful is to misunderstand it.*"

**Practical Considerations.**    Some noun phrases exhibit several types of vagueness. For example, a "long-lasting fame" exhibits both scalar vagueness (how long?) and subjective vagueness ("fame"). "Talented statesmen" is both subjectively vague ("talented") and quantitatively vague (how many statesmen? All? Some?).

Furthermore, vagueness cannot be decided on the noun phrase alone. It depends on the context in which the phrase is used. For example, the word "computer scientists" exhibits quantitative vagueness in the sentence "Computer scientists believe that AES encryption is safe" (because it is not clear how many share this belief), but no vagueness in "Computer scientists did not exist in the Middle Ages" (as there is no doubt about the quantity in this sentence).

Vagueness appears usually in non-named concepts. Named entities, such as the city of "Paris/ France", are not vague under most definitions of vagueness (except Loki's). Thus, the treatment of vagueness is inherently related to the treatment of non-named entities in natural language text [Rosales-Méndez et al., 2020, Paris and Suchanek, 2021]. However, not all non-named entities are vague. For example, "my garage" is not vague, even though it is not a named entity.

Vagueness is different from ambiguity [Zhang, 1998]: Ambiguity applies when a word has several distinct meanings, and only one of them is intended (as, e.g., "bank" for the financial institution and the river bank). A vague expression, in contrast, has one meaning but more than one interpretation, and these blend into each other.

Vagueness is different from underspecification. For example, the phrase "15 adults" is not vague – even though it does not specify who these people are. Phrases can nearly always be made more specific, by adding more details about the concerned entities. However, what matters for vagueness is not the amount of detail, but whether the phrase is concrete enough to appear in a law or contract. "15 adults" clearly is, whereas "many people" is clearly not.

With all of this in mind, we can now reconsider the example about the anti-tobacco sentiments from Section 1: Anti-tobacco sentiments are subjectively vague, because it is not clear when something counts as a "sentiment". "Many nations" exhibits quantitative vagueness, because the number of nations is in question. Finally, the "middle of the 19th century" shows scalar vagueness, because a given year can be more or less covered by that notion. The distinction between scalar and

| Vagueness | Number | Proportion | Examples |
|---|---|---|---|
| None | 1818 | 73.99% | "the company", "three children", "Paris", ... |
| Scalar | 148 | 6.03% | "long-running debate", "early history", "a large share", ... |
| Quantitative | 340 | 13.84% | "many of the ideas", "government bodies", "other media", ... |
| Subjective | 230 | 9.37% | "tensions", "high ethical standards", "sufficient interest", ... |

Table 1: Types of vagueness in our corpus

quantitative vagueness may seem redundant, but we will see in Section 5 that they are treated in different ways by existing approaches: scalar vagueness concerns a concept to which an individual instance can belong to a certain degree. Quantitative vagueness concerns a concept to which instances do or do not belong, although their number is unclear.

## 3. Case Study: Wikipedia

Even though there exist several works that define and classify types of vagueness (Section 2), we are not aware of a systematic quantitative study of the phenomenon in noun phrases. Therefore, we conducted a manual analysis of the vagueness in Wikipedia articles. Our choice is motivated by the fact that Wikipedia is a widely used standard reference, in both research and industry applications. We focus on the Wikipedia articles with highest quality (the "featured articles"), which generally do not contain weasel words, i.e., words that create the impression of importance while being de facto vague[1], or other types of vagueness deemed inappropriate by the Wikipedia community. In addition, if vagueness is present in the noun phrases alone, it is necessarily present at least as much or even more in all components of the sentences. Thus, our study constitutes a probable lower bound on the frequency of vagueness. We randomly choose one article from each of the 30 topics of the featured articles[2], and considered its abstract. We extracted the noun phrases with nltk[3], and manually verified them. We consider noun phrases that are sequences of nouns, adjectives, adverbs, determiners, and prepositions. Then we manually decide whether a noun phrase is vague, and if so, what type of vagueness applies. Overall, two computer scientists annotated 2457 noun phrases.

We also annotated each noun phrase with its plurality (plural vs. singular), its type of modifiers (adjectives, subordinate phrases, etc.), and its semantic class. Inspired by Yago 4 [Tanon et al., 2020], we used the top-level classes of schema.org combined with the top-level classes of BioSchemas.org, i.e. 11 classes in total.

We treated the document top down, and judged the vagueness of a word at the point where it appears, given all previous text. We did not treat the case where a later explication treats the vagueness that was encountered previously, as the word would still appear vague to human readers when they first encounter it. Besides, as explained in the previous section, annotators were allowed to tag the NPs with multiple types of vagueness. For example, "complex societies" is both subjective and quantitative.

Overall, our categorization was sufficiently clear and complete to categorize all encountered noun phrases. Inter-annotator agreement (Cohen's kappa) was 0.906,which is considered excellent.

---

1. https://en.wikipedia.org/wiki/Weasel_word
2. https://en.wikipedia.org/wiki/Wikipedia:Featured_articles
3. https://www.nltk.org/

| Class | Total | Vague | | Scalar | | Quantitative | | Subjective | |
|---|---|---|---|---|---|---|---|---|---|
| intangible | 546 | 215 | 39.38% | 49 | 8.98% | 68 | 12.46% | 125 | 22.9% |
| creativework | 457 | 120 | 26.26% | 15 | 3.29% | 89 | 19.48% | 36 | 7.88% |
| person | 367 | 71 | 19.35% | 8 | 2.18% | 50 | 13.63% | 18 | 4.91% |
| place | 269 | 38 | 14.13% | 12 | 4.47% | 25 | 9.3% | 6 | 2.24% |
| event | 250 | 58 | 23.2% | 36 | 14.4% | 18 | 7.2% | 13 | 5.2% |
| organization | 194 | 34 | 17.53% | 5 | 2.58% | 25 | 12.89% | 6 | 3.1% |
| action | 147 | 52 | 35.38% | 8 | 5.45% | 29 | 19.73% | 21 | 14.29% |
| product | 102 | 18 | 17.65% | 7 | 6.87% | 11 | 10.79% | 1 | 0.99% |
| taxon | 62 | 22 | 35.49% | 4 | 6.46% | 18 | 29.04% | 3 | 4.84% |
| biochementity | 58 | 10 | 17.25% | 4 | 6.9% | 6 | 10.35% | 0 | 0.0% |
| medicalentity | 5 | 1 | 20.0% | 0 | 0.0% | 1 | 20.0% | 1 | 20.0% |

Table 2: Proportion of noun phrases per semantic class

Examples of disagreement between annotators are "royal power" and "public interest", which were classified as scalar by one annotator and as subjective by the other. For the first example, the question is whether the power of a king can be measured on a common scale (i.e., scalar vagueness) or not (i.e., subjective vagueness). For the second example, the question is whether there is a scale that measures to what degree something is of public interest (i.e., scalar vagueness), or whether this is entirely subjective.

## 3.1 Vagueness in General

As Table 1 shows, a surprisingly large portion of noun phrases (26%) is vague. We detail the vagueness per semantic class in Table 2. As expected, the most vague class is intangible objects. These regroup many subjectively vague concepts such as beauty and ingenuity. Taxon noun phrases are not that numerous in our dataset, but are still vague 35% of the time. This is because they often appear in uncounted (and thus quantitatively vague) expressions such as "free-ranging herds" or "unique species of plants". The third-most vague class are actions: 35% of all actions are vague, mostly due to subjective vagueness as in "efforts" or "initial controversy" or quantitative vagueness as in "service initiatives" or "existing measures". The classes of person, place, organization, and product are rarely vague – despite their high frequency. This may be the reason why these classes have traditionally been targeted first by named entity recognition and anaphora resolution approaches. As expected, undetermined noun phrases are often vague (46%) because they often exhibit quantitative vagueness (as in "government bodies"). Noun phrases with an adjective are also very often vague (45%), which is mostly because these adjectives induce scalar vagueness (as in "a long-running debate") or subjective vagueness (as in "important train station"). Mass nouns are also vague 29% of the time, and this is because they tend to be subjectively vague (as in "fame"). Determined noun phrases are less likely to be vague (around 13%), and the vagueness then stems mostly from the adjective. Interestingly, there are 79 NPs with multiple types of vagueness (3.22% of all NPs, or 12.37% of vague NPs) and all combinations of vagueness exists in our dataset.

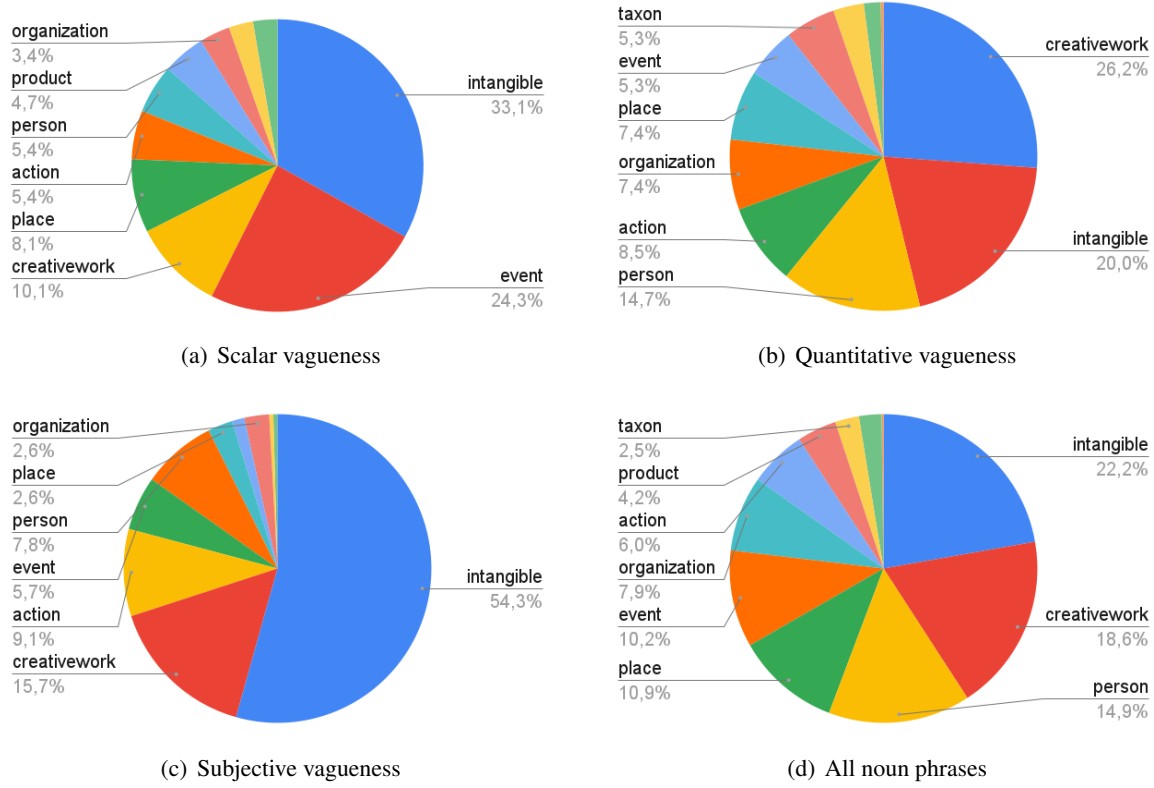

(a) Scalar vagueness

(b) Quantitative vagueness

(c) Subjective vagueness

(d) All noun phrases

Figure 1: Proportion of semantic classes in each type of vagueness – and among all noun phrases (vague and non-vague)

## 3.2 Scalar Vagueness

Scalar vagueness is the least frequent vagueness type, with only 6% as shown in Table 1. Scalar vagueness is usually (in 66% of the scalar vague noun phrases) induced by an adjective. This finding is in line with previous work, which found that vagueness is commonly associated with the usage of terms such as "minimal", "later", "appropriate", etc [Rosadini et al., 2017]. In many cases, the scalar vague noun phrases relate to time and space, as in "a long-running debate" or "large interior sites". Indeed, 24% of the scalar vague noun phrases are events (Figure 1). These are mainly imprecise periods such as "the late Cretaceous period", "the early 20th century" or "a short stint with the Boston Braves". The main other class of scalar vagueness is intangible objects (such as "a small cost" or "any large economic bubble").

## 3.3 Quantitative Vagueness

Quantitative vagueness is the most frequent type of vagueness, affecting 14% of the noun phrases (Table 1). As expected, plural noun phrases are more likely to be quantitatively vague than any other type of vagueness (44% of plural noun phrases are quantitatively vague), and vice versa, quantitative vagueness concerns almost exclusively (97%) plural noun phrases. The largest portion of these

(67%) is made up by undetermined nouns. This was expected, because an undetermined plural noun is almost always quantitatively vague, unless it is accompanied by a concrete quantity ("fans" is quantitatively vague, "5 fans" is not). 19% of quantitatively vague noun phrases come with a (vague) quantity, such as "several", "many", or "few". Naturally, mass nouns are not so much represented (7% of the quantitatively vague phrases), because they cannot appear in numbers. However, they can still appear in portions (as in "much of the water").

From the class perspective (Figure 1), we see that a large portion of quantitatively vague noun phrases (26%) concerns creative works. This is because our definition of creative works includes anything that can be conceived and created by the human mind, such as "high scholastic and ethical standards" (a phrase that is also subjectively vague), "community service programs", or "current policies", and these concepts appear frequently in plural in our corpus.

### 3.4 Subjective Vagueness

Finally, 9% of the noun phrases are subjectively vague (see Table 1). These are nouns such as "tensions", which are reasonably clear to the reader, but too vague to appear in a law or a contract. The same goes for "sufficient interest", "his greatest fame", "the sport's popularity", and other such terms. Indeed, half of the subjectively vague phrases concern such intangible objects (Figure 1). Creative works are also quite frequent (at 16%), and this is again because our definition includes concepts such as "fantastic elements", "a realistic portrayal", or "its finest works".

In many cases (62%), subjective vagueness comes in noun phrases with adjectives, and quite often it is the adjective that makes the noun phrase subjectively vague (as in "the fanciful manner", or "a transformative era"). Mass nouns make up 31% of the subjectively vague noun phrases ("royal power", "his international reputation"), and this is because they are often intangible concepts, which in turn are often subjectively vague.

We emphasize again that this vagueness is not a default of Wikipedia: vague expressions are often the most effective means to communicate a message without being overly specific [Lim and Wu, 2018]. In many cases, it may just be impossible to use a more specific noun phrase, since the details or the boundary conditions are unknown. Thus, vagueness appears to be a useful, and possibly necessary feature of natural language. Our study consequently confirms that the phenomenon is rather frequent. Therefore, it is interesting to see how current computational approaches deal with vagueness – which is what we will now detail.

## 4. Detecting vagueness

We now come to approaches that detect vagueness automatically in natural language text. The CoNLL shared task on detecting uncertainty [Farkas et al., 2010] focused on detecting hedges and vagueness in a biomedical corpus and in Wikipedia. Indeed, Wikipedia has developed an extensive guideline of what constitutes *weasel words*. Debnath and Roth [2021] investigated the ability of a neural model to distinguish between vague and non-vague versions of instructions in WikiHow[4] edits. They studied vagueness based on the change in the main verb in the original and revised version of an instruction. Sinha and Dasgupta [2020] focus on the vagueness of verbs and adverbs, as in "The gas leakage probably caused the fire accident". All of these works study vagueness on the sentence level, not on the noun phrase level.

---

4. https://wikihow.com

Most works on the vagueness of noun phrases focus on specific application domains: Alexopoulos and Pavlopoulos [2014] detect vague definitions in an ontology about citations. For this purpose, the authors manually created a dataset of 1000 vague adjective definitions, and 1000 non-vague definitions in WordNet. The vagueness corresponds mostly to subjective vagueness and scalar vagueness in our classification. They then trained a classifier on this dataset, and applied it to the definitions in their ontology. The results show that a simple bag-of-word classifier achieves a good accuracy, suggesting that the presence of certain words (such as "usual" or "intermediate") indicates vagueness rather well in this ontology.

Rosadini et al. [2017] study quality defects in requirement specifications in the railway domain. One of these defects is the vagueness of the specifications. The work uses manually defined NLP Patterns to identify quality defects, and a set of manually defined phrases (such as "as much as possible") to detect vagueness – mostly of the scalar type. The authors note, however, that they encountered difficulties in defining phrases that identify vagueness with a high accuracy.

Reidenberg et al. [2016] and Liu et al. [2016] study vagueness in the context of privacy policies. The first work conducted a user study to identify vague formulations in privacy policies. This led to the collection of 40 words and phrases that indicate vagueness. Many (such as "generally", "might", etc.) concern the sentence level, and among those that concern the noun phrase level, we find indicators of both quantitative vagueness (such as "some") and subjective vagueness (such as "appropriate"). The second work builds on the first. The goal is to learn a vector representation of words that can perform well in two tasks: predicting the next word given its preceding words, and predicting whether or not a word is vague. When this model is trained with a high weight on the error of the vagueness prediction, it achieves a high accuracy in this task very quickly. However, the findings also suggest that there exist so many different ways of expressing the same vague concept that it seems impractical to compile an exhaustive list of them. Lebanoff and Liu [2018] asked crowd workers to annotate phrases in privacy policies by vagueness scores between 1 and 5. This exercise identifies vagueness both on the level of noun phrases and on the level of sentences (as indicated by sentence-level adverbs or verbs) – without a distinction of the category of vagueness. Then the authors train two types of classifiers to predict vagueness: the first, an LSTM, is context-aware; the second, a feed forward network on the word embeddings, is not. The results show that the context-dependent classifier performs much better than the other one. This confirms our intuition that a phrase is vague only in its context.

## 5. Reasoning with Vagueness

Most efforts to model vagueness have been put into solving the Heap Paradox [Roschger, 2014]. Several proposals have been developed in the fields of philosophical logic [Fine, 1975, Lewis, 1979, Soames et al., 1999, Williamson, 2002] and linguistics [Seuren, 1973, Cresswell, 1976, Lewis, 1976, Klein, 1980, Kyburg and Morreau, 2000, Barker, 2002, Kennedy, 2007, Kamp, 2013]. Here, we concentrate on methods that allow for a computational approach to vagueness. Indeed, we believe the pragmatic effect is mostly an issue for modeling approaches than humans since vagueness is a natural feature of our communications.

Generalized quantifiers [Mostowski, 1979, Perlindström, 1966, Barwise and Cooper, 1981, Tu and Madnick, 1997] (GQ) are generalizations of the standard first-order logic quantifiers $\forall$ and $\exists$. GQs allow modeling and reasoning with sentences such as "More than half of John's arrows hit the target" (i.e. quantitative vagueness in our classification). Fragments of first-order logic or modal

logic can be extended with such quantifiers. Interestingly, validity is decidable for most of these extensions, which distinguishes them from First Order Logic.

As an example, let us consider the sentence "*Most cities in Taiwan are major cities*" and the generalized quantifier $Q_{Most}$, which stands for "most" and means that $|A \cap B| > |A - B|$ is true with $A$ being the set of Taiwanese cities and $B$ the set of major Taiwanese cities. Then, $Q_{Most}$ can be embedded into description logic with a new operator $\rightarrow_{Q_{Most}}$[Tu and Madnick, 1997], and the quantitative vagueness in "*Most cities in Taiwan are major cities*" can be expressed as:

$$City(x) \wedge Located(x, Taiwan) \rightarrow_{Q_{Most}} is(x, Major)$$

Fuzzy sets [Zadeh, 1965] are sets whose elements have varying degrees of membership. These can be used to model scalar vagueness, as in "tall women". Fuzzy Logic has given rise to several approaches in NLP. Zadeh [1975], Hersh and Caramazza [1976], Zadeh [1983], represent natural language concepts as fuzzy sets, and define operators on them. These operators take the form of adverbs, negative markers, and adjectives, and they allow modeling phrases such as "quite large", "sort of short", "not very old". With the appropriate membership functions, one can also make sure, e.g., that a person of 190cm belongs to the fuzzy set *tall person*, and this allows for reasoning.

Straccia [2008] add Zadeh's formulation of Fuzzy Logic [Zadeh, 1965] to the $\mathcal{ALC}$ Description Logic in order to model scalar vagueness in ontological concepts. The formalism allows for both fuzzy terminological (T-Box) and fuzzy assertional (A-Box) knowledge, but not for fuzzy modifiers such as "very", "more or less", or "slightly". For more details about fuzzy Description Logics, [Ma et al., 2013] proposed a complete overview.

Alexopoulos et al. [2014] and Jekjantuk et al. [2016] proposed the Vagueness Ontology, a meta-ontology to identify and derscribe vague entities and their vagueness-related characteristics. This allows users to annotate vague classes or properties with descriptions that reduce the possible interpretations, so as to improve their comprehensibility and shareability. For example, the user can specify the type of vagueness (scalar or quantitative), as well as the dimensions (in the case of quantitative vagueness). The user can also model the way vagueness and its characteristics propagate when defining more complex OWL axioms (such as conjunctive classes) by means of formal inference rules and constraints.

Lassiter and Goodman [2017] proposed a Bayesian approach to model scalar vagueness in adjectives such as "tall", "heavy", and "happy". They mention that this approach could be extended to other scalar expressions (such as some verbs or quantifiers). Sentifiers [Setlur and Kumar, 2020], too, is a system to model scalar vague modifiers such as "worst" or "more". The system allows visualizing vague concepts, e.g., by displaying a map when the user asks for the location of "unsafe" places. For this purpose, the algorithm maps the vague modifiers to numerical attributes using word co-occurrence. Then, sentiment analysis is used to determine the filter ranges applied to the attributes. A recursive neural network determines the sentiment class from *very negative* to *very positive*. Finally, these features are used to compute a range on the scale of vagueness.

## 6. Vision

We have seen that vagueness is an important and frequent feature of natural language, and that several existing works tackle different aspects of the phenomenon. However, if we want to take vagueness into account in knowledge bases and their applications such as chatbots, assistants, or

question answering, we would need an end-to-end handling of vagueness in natural language text. The first step of such a pipeline would be the detection and categorization of vagueness. We believe that the existing approaches could be extended, and trained on our dataset to this end. To illustrate this, we conducted an experiment using a simple Ridge classifier with built-in cross-validation for multinomial models on our corpus. Using Scikit-Learn[5], we obtained an accuracy of 0.78 when classifying noun phrases according to their vagueness type(s), i.e. scalar, quantitative, subjective or not vague. This approach can obviously be improved (for example with more features, more training data, or more advanced machine learning machinery), and we thus believe that it is feasible to detect and categorize vagueness in natural language texts.

The more tricky part is the modeling of vagueness in a knowledge base. Scalar vagueness seems to be the easiest category of vagueness to model since all approaches under consideration (except generalized quantifiers) can handle it. Quantitative vagueness can be modeled only with generalized quantifiers. One possible way forward would thus be to integrate the work on fuzzy sets [Straccia, 2008] and generalized quantifiers [Tu and Madnick, 1997] into description logics, so as to represent both scalar and quantitative vagueness. Alternatively, quantitative vagueness could be modeled with anonymous instances of ad-hoc classes and axioms, or with ad-hoc classes [Rosales-Méndez et al., 2020, Paris and Suchanek, 2021]. In our example, "many nations" could become an anonymous subclass of the "nation" class. Then, an axiom could say that all instances of that class (whatever they are) harbored anti-tabocco sentiments. Qualifiers such as "most" would have to be modeled by appropriate axioms (saying, e.g., that the number of instances with a certain property is larger than the number of other instances). Tackling scalar and quantitative vagueness would cover already two thirds of the cases.

Subjective vagueness is certainly the hardest nut to crack. One possibility is to annotate every subjectively vague phrase with the person who employed it. For example, the sentence "The company demands that its employees follow high ethical standards" would have to be rewritten as "...follow the standards that it considers high ethical standards". In this way, some limited type of reasoning would still be possible (deducing, e.g., that there are standards that the employees have to follow), and different entities can consider different standards highly ethical. However, such an approach would need to model statements that are true only in certain contexts – which is a research topic on its own [Suchanek, 2020].

## 7. Conclusion

In this position paper, we have studied the phenomenon of vagueness in noun phrases. We have identified three types of vagueness (scalar, quantitative, and subjective), and we have found that a substantial portion of noun phrases in our Wikipedia corpus exhibit vagueness of one type or the other. We have thus concluded that vagueness is a frequent phenomenon, even in curated text. We have surveyed the literature on the topic, and found that while there are several approaches to model and reason on vagueness in general, the approaches have so far been limited to application-specific scenarios or to specific types of vagueness. Vagueness thus remains a phenomenon that is frequent, but yet underexplored.

All our annotations and statistics are publicly available at our GitHub page[6].

---

5. https://scikit-learn.org/stable/modules/generated/sklearn.linear_model.RidgeClassifierCV.html
6. https://github.com/dig-team/vagueness_project

## Acknowledgments

This work was partially funded by the grant ANR-20-CHIA-0012-01 ("NoRDF").

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
