# OpenReview forum: "The Vagueness of Vagueness in Noun Phrases"
_AKBC.ws/2021/Conference — AKBC 2021_

### Official Review · Reviewer_vVo1 · 2021-07-20
**A well-crafted and thought-provoking study of vagueness with limited outcomes.**

**Rating:** 6
**Confidence:** 4

**Review:**

# General comments
In this paper the authors discuss the problem of vagueness in NLP, present a new categorisation scheme for vague noun phrases (NPs) and demonstrate through a case study annotation of Wikipedia passages that their categorisation is intuitive enough to lead to high inter-annotator agreement.

The paper is a well-crafted resource for reviewing the literature in the field and the case study (and subsequent analysis) is well designed and potentially useful as a resource in itself. My only concern is that if we consider this exclusively as a position paper, other than "vagueness is still vague" it's not clear what the position is. I would have liked to see some more concrete suggestions in the final section on how the recommended categorisation scheme advances the research in the field, especially when it comes to practically representing vagueness in knowledge graphs and other NLP applications.


# Specific comments/questions
In the Definition paragraph of Section 2, it's not clear why "head" is not vague under the vagueness doctrine of American constitutional law. If this a hypothetical example, I would imagine a similar argument could be made for "heap" (where a court ruling or a law would define "heap" in terms of e.g. weight or countable items). If it is a concrete legal example, I would like to see some more details as to why "head" was able to defined but a term like "heap" wasn't.

The example of "beauty" is used to justify the rejection of previous categorisations but the reasons for that rejections are themselves vague. It's not clear why neither the "referent unclear" or the "overlapping of disjoint but overlapping concepts" definitions fit the type of vagueness described by "beauty". It would be helpful to see what types of disagreements or coverage gaps appeared during the annotation effort when using one of the existing categorisations.

The terms "meaning" and "interpretation" used in the paragraph "Practical Considerations" should be defined.

Is the proposed categorisation system adhering to either definition given in Section 2? In particular the vagueness doctrine seems to provide a very useful functional definition, even for the example used in the first section. I can imagine for instance a word like "sentiment" being operationalised for a specific legal case. In addition, it would be an interesting exercise to annotate patents or other legal documents with the same annotation scheme applied to Wikipedia and compare the prevalence of vague words in the two fields.

A few questions on the Wikipedia annotation:
- Are named entities (proper nouns) included in the set of NPs considered for annotation? If not, how were they identified and removed?
- What tool was used for identifying the different PoS tags used the NP defintion?
- How many semantic classes were used in total?
- Were annotators allowed to choose more than one type of vagueness?

The result of the baseline classifier are already higher than the reported IAA (0.78 vs 0.744). This implies that the dataset is already "solved", unless there were efforts to ensure higher agreement post-hoc and the adjudication of disagreements was itself consistent.

The work by Kees van Deemter "Not Exactly: In praise of vagueness" is a great reference that needs to be included, not only for its coverage of different definitions and categorisations, but also because of the discussion of the practical aspects of dealing with vagueness in AI.


# Minor comments
"Weasel word" is used (Section 3) before being formally defined (Section 4)

---

> ### Author Response · Authors · 2021-07-29
> **Thanks for your review and feedback! We have uploaded a revised version of the paper with your suggestions.**
>
> - "*I would have liked to see some more concrete suggestions when it comes to practically representing vagueness in knowledge graphs and other NLP applications.*" Section 5 presents several formalisms to this end. Section 6 discusses how these could be combined. However, our work is a position paper, whose goal is to entice the community to look into the problem. To this end, we provide a survey of existing building blocks, but we explicitly keep the solution open-ended.
> - "*I would like to see some more details as to why "head" was able to defined but a term like "heap" wasn't.*" The word "head" appears in legal contracts such as insurance policies to protect people from [head injuries](https://www.google.com/search?q=%22head%22+accident+insurance). Hence, the "head" is not vague. The word "heap" is not vague either, if it is accompanied by measures of size or weight (as you suggest). Without these, it is vague, and could not appear in a contract.
> - "*It would be helpful to see what types of disagreements or coverage gaps appeared during the annotation effort when using one of the existing categorizations.*" We did not encounter any coverage gaps. Disagreements arose mainly when annotators forgot that a noun phrase has a subjectively vague component in addition to other types of vagueness.
> - "*The terms "meaning" and "interpretation" used in the paragraph "Practical Considerations" should be defined.*" The term "meaning" stands for what the sender (the writer) wants to convey, while "interpretation" concerns what the receiver (the reader) understand from the text. The paper has been updated.
> - "*it would be an interesting exercise to annotate patents or other legal documents with the same annotation scheme*" Indeed, this would be a very interesting exercise, e.g., on the privacy corpus of [Reidenberg et al., 2016].
> - "*Are named entities (proper nouns) included in the set of NPs considered for annotation? If not, how were they identified and removed?*" Yes, named entities have been manually identified and kept. We added an example to the paper.
> - "*What tool was used for identifying the different PoS tags used the NP definition?*" We used **nltk** to tag the text.
> - "*How many semantic classes were used in total?*" **11 classes** were used. The paper has been updated accordingly.
> - "*The result of the baseline classifier are already higher than the reported IAA (0.78 vs 0.744). This implies that the dataset is already "solved"*" The inter-annotator agreement (IAA) is not a percentage, and thus cannot be compared to the accuracy.
> - "*The work by Kees van Deemter "Not Exactly: In praise of vagueness" is a great reference that needs to be included*" Thank you for the pointer! We now cite it to cement the observation that vagueness is both unavoidable and useful.
> - "*"Weasel word" is used (Section 3) before being formally defined (Section 4)*" You're right, the error has been corrected.

---

> > ### Comment · Reviewer_vVo1 · 2021-07-30
> > **Thanks for the clarifications**
> >
> > I would like to thank the authors for addressing my questions and adding some clarifications regarding concepts.
> >
> > Regarding the vision and motivation of the paper, I am still not entirely convinced by the author's response. I agree that the KB community has largely ignored vagueness, but that's not because they aren't aware of its existence, but rather that it is very hard to address concretely and at scale. The current work does indeed provide a valuable contribution in this area by systematising the categories and providing quantitive results (and a useful resource) but I don't see the thesis of the position paper itself as the main contribution.
> >
> > Finally a minor clarification on the following:
> >
> > > The inter-annotator agreement (IAA) is not a percentage, and thus cannot be compared to the accuracy.
> >
> > This is merely a technicality. Cohen's kappa is chance-corrected agreement which itself is just a percentage. My point was if two experts (or any humans) agree only about 74% of the cases, this means that the highest possible score achievable by a human (against a fully-adjudicated version of the dataset) is 74%.

---

### Official Review · Reviewer_ygk9 · 2021-07-21
**fascinating exploration of vagueness in noun phrases, with many open questions and some gaps in the conceptual framework**

**Rating:** 6
**Confidence:** 3

**Review:**

# General comments
This paper presents a fascinating attempt to define "vagueness" as it applies to noun phrases, and to annotate it and model it in natural language text.  The authors attempt to arrive at a working definition of NP vagueness and then annotate noun phrases using the simple schema they develop.  I have two main concerns: the first is that I found some of the definitional parts of the paper somewhat lacking in rigor, and the second is that I am not certain what the main position of the authors is, other than "vagueness is rampant in natural language" and "researchers have not explored it enough".  I *suspect* the position is that we need to do a better job of modeling vagueness so that our knowledge representations and our ability to answer questions about knowledge expressed in natural language form can be improved, but these points were things I could only surmise but were not exactly made explicit: they were raised in the introduction but not really fleshed out in the "Vision" or conclusion sections.

# Specific comments
* It is a little too strong to say that the use of any vague noun phrases makes a sentence "inaccessible to current knowledge extraction methods".  Perhaps the authors are thinking only about KB's/KG's that consist of named entities and facts related to those entities?
* We can and should differentiate with NP's that are vague in their current sentence, but which could be made more precise, either by understanding the greater discourse or social context, and/or by further research.  In the example sentence "An anti-tobacco sentiment grew in many nations from the middle of the 19th century", we can imagine that we could reasonably ask "in which nations did anti-tobacco sentiment rise after 1850?"
* The definition of ambiguity seems overly rigid.  A lot of work in lexical semantics and word sense disambiguation (yes, that was an intentionally vague phrase) has looked at resolving the ambiguity of even closely related definitions of words; see, e.g., the definitions of the verb "cut" in WordNet.
* It is somewhat odd to group "subjectivity" into "vagueness".  Subjectivity just means that the truth value is dependent upon the observer, but not that the concept is lacking in precision for whatever reason.  It certainly does mean that any "knowledge" that we might want to extract should probably be attached to that observer, as the authors note near the end of Section 6.
* Similar to the above, it is somewhat odd to conclude that "beauty" is "clearly vague".  I think it is that beauty is both a subjective concept *and* it is complex and/or hard to define.
* Some questions:
  * How much of the vagueness observed with noun phrases was resolvable when reading the entire document?  Natural language derives much of its expressive power and compactness due to its pervasive use of ambiguity and vagueness, coupled with our ability to come up with helpful interpretations based on many different types of context to resolve ambiguities and vague expressions.
  * You mention how a phrase can exhibit multiple forms of vagueness.  How often does that happen, and with what types?
  * In your "Vision", you mentioned that we should have a better, end-to-end treatment of vagueness.  Have you considered extending a question-answering dataset so that it includes questions where the answers—or support for the answers—lie in sentences with vague noun phrases?

---

> ### Author Response · Authors · 2021-07-29
> **Thanks for your review and feedback! We have uploaded a revised version of the paper with your suggestions.**
>
> - "*It is a little too strong to say that the use of any vague noun phrases makes a sentence "inaccessible to current knowledge extraction methods". Perhaps the authors are thinking only about KB's/KG's that consist of named entities and facts related to those entities?*" Yes, we are referring mainly to knowledge base building. The paper has been updated.
> - "*We can and should differentiate with NP's that are vague in their current sentence, but which could be made more precise*" Indeed, but we believe that this is a goal in a second step. In this paper, which is a first step, we circumscribe the problem by reviewing the literature in terms of definitions, detection, modeling, and we conduct a quantitative study.
> - "*The definition of ambiguity seems overly rigid.*" You are right. We took the definition from [Zhang, 1998], but we have now opted for a loser definition.
> - "*Have you considered extending a question-answering dataset so that it includes questions where the answers—or support for the answers—lie in sentences with vague noun phrases?*"
> This is certainly something we want to try!

---

### Official Review · Reviewer_vazn · 2021-07-23
**Fun discussion and study of vagueness in NPs. Corpus limited in size and could be better defined.**

**Rating:** 6
**Confidence:** 3

**Review:**

This paper presents a discussion of vagueness of noun phrases along with a categorization of types of vagueness, and a corpus study that measures their prevalance in English Wikipedia. There ais a nice related work section that discusses efforts to identify and reason with vagueness in NLP, with a focus on formal reasoning systems. The authors suggest that their corpus could e a useful resource in creating detectors of vague NPs in a system that then has special reasoning capabiilties designed to deal with this vagueness.

I enjoyed reading the paper and I think that discussions of vagueness, underspecification, and ambiguity are becoming increasingly important as the reasoning capabilities of our NLP systems are reaching the point where these phenomena are important. However, I also think that the current paper would be stronger with a more formal definition of the process by which the two annotators identified vague NPs, in particular with relation to the effect of context. I'd also like to see a discussion of the pragmatic effect of vague NPs on a listener --- i.e. does it actually affect the listener's interpretation or is it mostly just an issue for set-theoretic modeling approaches.

The authors do mention the importance of context, but do not tel us how much context the annotators consider. If the annotators are considering anything less than the full document context, I would like to know to what extent the different forms of vagueness are reducible through the consideration of more context. Similarly, it would be nice to know to what extent the vagueness can admit signicantly different interpretations to a reasonable reader.

Strengths:
- Interesting discussion of vagueness, which is distinct from other types of ambiguity often discussed in the field.
- Nice overview of related work from adjacent fields.
- Enjoyable to read.

Weakensses:
- Definition of annotation process could be more formal. In particular w.r.t. amount of context considered and importance of pragmatics.
- Corpus is small and likely to be of limited utility.

---

> ### Author Response · Authors · 2021-07-29
> **Thanks for your review and feedback! We have uploaded a revised version of the paper with your suggestions.**
>
> - "*Corpus is small and likely to be of limited utility.*" We disagree here. Our corpus covers more than two thousand NPs and has proven to be sufficient to train a classifier to detect and classify vagueness (see Section 6). It thus serves its purpose. That said, future work can indeed extend the corpus to more annotations.

---

### Author Response · Authors · 2021-07-29
**Thanks for your review and feedback! We have uploaded a revised version of the paper with your suggestions.**

Reviewer vVo1 = R1

Reviewer ygk9 = R2

Reviewer vazn = R3

We thank the reviewers for their thoughtful remarks. We are encouraged they found our study about vagueness well-crafted (R1), fascinating (R2) and fun (R3). We are glad they found our categorization intuitive, the case study well designed and useful (R1), and the problem of vagueness important (R3). We will now reply to each question.

**@R1,R2: What is our vision?** Our position is that vagueness is currently ignored when it comes to knowledge bases and their applications such as question answering systems, and that vagueness should be taken into account. We have made this clearer in the "vision" section. More specifically, we define a problem, categorize its instances, analyze its nature, provide a dataset, discuss the approaches that have done something similar, and call for the community to look into the problem.

**@R1,R2: Why is beauty vague?** Even a single observer considers that beauty applies to a certain degree ("Alice is more beautiful than Bob"). Yet, there is no consensus on how to measure this degree (someone else may find Bob more handsome than Alice). This makes the concept subjectively vague. In general, we consider vague any noun phrase that cannot appear in a contract, and beauty qualifies here.
Lassiter and Goodman [2017] see just scalar vagueness and “everything else”, under which they also group nouns such as “bird” (presumably because of a lack of precision). Bennett [2005] defines a form of vagueness as the "overlap of disjoint but overlapping concepts". For example, Paris in France is both a city and a department, and both objects are the same w.r.t. geography, but are different from a legal perspective. Again, this definition doesn't apply to beauty. Hence the need to introduce a new category, which is what our work does. We find that this category is actually very frequent: It concerns a large number of noun phrases that carry an opinion (like "sophisticated" or "fame"), and/or that cannot be assessed objectively (like "power" or "great ability").

**@R2,R3: In which context do you analyse vagueness?** We treat each document top down, and judge the vagueness of a word at the point where it appears, given all previous text. We did not treat the case where a later explication treats the vagueness that was encountered previously, as the word would still appear vague to human readers when they first encounter it.

**@R1,R3: What about multiple forms of vagueness in a noun phrase?** Annotators were allowed to tag the NPs with multiple types of vagueness. For example, "complex societies" is both subjectively vague and quantitatively vague. Our paper discusses this in Section 2, but we also updated the use case section to be clearer. There are 79 NPs with multiple types of vagueness (3.22% of all NPs, or 12.37% of vague NPs). All combinations of vagueness exists in our dataset.

The paper has been updated to take into account all of these points.

---

### Decision · Program_Chairs · 2021-08-17

**Decision:**

Accept

**Comment:**

This paper presents a corpus of noun phrases annotated with "vagueness", with "vagueness" itself split into multiple types (e.g., subjective versus quantitative). The paper is particularly relevant to AKBC because KBs often assume categorical, unambiguous entries, which vague NPs can thwart. The reviewers' overall sentiment was positive. The paper contributes an interesting dataset and its positioning in the literature should inspire good discussion. We also urge the authors to incorporate the reviewer's comments into the camera-ready version.